# Tuberculosis yield among contacts of non-pulmonary bacteriologically confirmed index TB patients in the urban setting of central Uganda

**Herbert Kisamba**[1◉], **Nicholas Sebuliba Kirirabwa**[1◉]*, **Kenneth Mutesasira**[1], **Seyoum Dejene**[2], **Abel Nkolo**[1]

**1** USAID Defeat TB Project, University Research Co. LLC, Kampala, Uganda, **2** USAID Uganda, Kampala, Uganda

◉ These authors contributed equally to this work.
* nkirirabwa@gmail.com

**Data Availability Statement:** All relevant data are within the paper and its Supporting information files.

## Abstract

### Background

The World Health Organization (WHO) recommends systematic and active investigation of TB contacts. However, lower priority is given to contact investigation among other non-pulmonary bacteriologically confirmed (PBC) cases; it thus contributes to the scarce information on the yield of TB among contacts of index TB patients without microbiological confirmation (non-PBC patients). This study therefore aimed at establishing the yield of TB among contacts of PBC and non-PBC index TB patients in the urban setting of central Uganda.

### Methods

We abstracted data from the Uganda national TB contact investigation registers present at 48 health facilities for the period January 2018 to August 2020. The screening yield for both PBC and non-PBC, timing of TB diagnosis among contacts were determined. Logistic regression was used to examine predictors for diagnosing contacts as non PBC TB patients.

### Results

From January 2018 to August 2020, 234 persons were diagnosed with TB from a total of 14,275 contacts traced for both PBC and non-PBC TB index patients at 48 facilities. Of these, 100(42.7%) were contacts of non-PBC index patients. TB screening yield was higher among contacts of non PBC 100(2.0%) compared to 134(1.4%) among contacts of PBC index patients. For both groups, over 80% of their contacts were diagnosed with TB within 3 months from the day of TB treatment start of the index case. On multivariate logistic regression the only predictor for diagnosing contacts as non PBC TB patients was age under 15 years (adjusted odds ratio [aOR] 7.53, 95% CI [3.27–17.3] p = <0.05).

**Funding:** This work was funded by the Unites States Agency for International Development (USAID)'s Defeat TB project Under Cooperative agreement No. AID-617-A-17-00003 through financial support. The funder reviewed and took the decision to publish.

**Competing interests:** The authors have declared that no competing interests exist.

## Conclusion

The yield of TB among contacts of non-PBC index case is nearly the same for contacts of PBC index cases and most contacts were diagnosed with TB disease during the intensive TB treatment phase of the index case. There was no association between the type of TB (PBC, non-PBC) disease diagnosed in the contacts, and that of index TB patients. To improve TB case-finding, emphasis should be placed on contact investigation for household and close contacts of all other index cases with pulmonary tuberculosis regardless of whether PBC or non-PBC during the intensive phase of treatment.

## Background

Investigation of contacts of patients with tuberculosis (TB) was a priority for TB control in high-income countries [1] until the World Health Organization (WHO) recommendation for systematic and active investigation of TB contacts targeting both low- and middle-income countries. This aimed to contribute to early identification of active TB in low- and middle-income countries [2].

Despite many recent advances in the diagnosis of TB, in 2021 the Uganda National TB and Leprosy Programme (NTLP) indicated a TB notification of 69,162 of the targeted 82,341 incident TB cases. This implies that about16% of the estimated incident TB cases in Uganda are missed each year [3]. The estimated TB incidence of 200/100,000 population of Uganda [4] is likely to decrease when most of the missed cases are identified and successfully treated to stop further transmission.

In 2013, the Uganda NTLP started offering household TB contact investigation as a routine public health service in Kampala [5]. The Ministry of Health (MOH)-NTLP, in 2019 through support from the Global Fund and implementing partners developed and launched an active case-finding (ACF) toolkit for improving the quality of TB care, to find the missing people with TB and successfully treat them. The ACF tool kit guides health facility teams to identify and analyses gaps in TB care and develop and test changes to address gaps in delivery of quality TB care at health facilities. As part of access to TB services, the ACF toolkit focuses on screening all contacts for active TB carried out through TB contact investigation.

Contact investigation involves the systematic evaluation of the contacts of known TB patients to identify active disease or latent TB infection (LTBI). It is one of several ACF strategies that have been proposed to increase case detection. It contributes to early identification of active TB, thus decreasing its severity and reducing transmission of *Mycobacterium* tuberculosis to others, and identification of LTBI, to allow preventive measures. However, TB contact investigation is rarely and inconsistently carried out in resource-limited settings [6]. There has been limited information on the yield of TB resulting from contact investigation among contacts of non-PBC index TB patients. Since lower priority is given to contact investigation among other non-PBC cases, it thus contributes to the scarce information on the yield of TB among contacts of index TB patients without microbiological confirmation (non-PBC patients). This study therefore aims at comparing the yield of TB among contacts of PBC and non-PBC (both pulmonary clinically diagnosed [PCD] and extra pulmonary [EP]) index TB patients.

## Methodology

### Study setting

The study data was collected from selected high-volume TB diagnostic and treatment units (DTUs) in two urban districts of Kampala and Wakiso, located in the central region of Uganda. The two districts neighbor each other with Wakiso partly encircling Kampala, the nation's capital, and largest city. The combined population of the two districts is approximately 4.38 million with Wakiso having the highest population at 2.73 million, and Kampala with 1.65 million [7]. Comparable to many urban districts in sub-Saharan Africa, the two districts have predominant urban populations and mobile populations living in slums with high population density, poor ventilation, and high TB transmission [8]. TB screening, diagnostic, and treatment services are available and provided free to the user in DTUs.

We considered PBC patient as one from whom a biological specimen was positive by smear microscopy, culture or WHO approved rapid diagnostics like Xpert MTB/RIF. The non PBC included PCD and EP, where PCD referred to any patient who did not fulfill the criteria for bacteriological confirmation but was diagnosed with active TB by a clinician or other medical practitioner and decided to give the patient a full course of TB treatment. Extra pulmonary referred to any bacteriological confirmed or clinically diagnosed patient of TB involving organs other than the lungs.

### Study procedures

This was a descriptive study with an analytical component using data routinely collected from 48 health facilities within Kampala and Wakiso that are supported by the USAID Defeat TB project. We abstracted data from the project electronic database that captures information from national TB contact investigation registers at the health facilities for the period January 2018 to August 2020. For those contacts diagnosed with TB within this period, their records were reviewed to obtain data on patient characteristics, *(age*, *sex*, *HIV status*, *TB type*, *relation to index*, *diagnostic test used*, *common symptoms among the diagnosed TB contacts)*, number of presumptive TB cases found, the number investigated for TB and the number diagnosed with TB, timing of TB diagnosis among contacts from the day of TB treatment start of the index case. This period was compared to intensive and continuous phases of TB treatment of the Index patient. The first phase which lasts two months of the standard six month course of treatment represent the intensive phase while the second phase that lasts four months is the continuous phase.

### Data management and analysis

Data was imported from the project electronic database, cleaned, coded, and maintained using Microsoft Excel 365. Data was exported and analyzed using STATA/IC version 13.1. For patient characteristics, descriptive statistics were tabulated for frequencies and percentages. To construct the TB diagnostic cascade, we calculated the proportion of presumptive TB contacts (a contact that has symptoms or signs suggestive of TB identified out of all contacts traced). We then calculated the proportion of contacts with presumptive TB tested for TB out of all contacts with presumptive TB identified, and finally the proportion of TB cases diagnosed out of all presumptive TB contacts tested. The screening yield for both PBC and non-PBC was calculated (the proportion of contacts diagnosed out of all those screened for TB). We calculated timing of TB diagnosis among contacts as the interval from the day of TB treatment start of the index case to contact TB diagnosis. Logistic regression was used to examine factors associated with diagnosing contacts as non PBC TB patients. Factors which had p-value ≤0.2 at bi-

variable analysis were entered into a multivariable logistic regression model. Variables with p-value ≤0.05 on multivariable regression were considered as statistically significant factors associated with diagnosing contacts as non PBC TB patients.

### Ethics statement

The need for informed consent for this study was waived by Joint Clinical Research Centre (JCRC) research and ethics committee. This granted the use of routinely collected data by Defeat TB project. All experimental protocols in the study were approved by Joint Clinical Research Centre (JCRC) research and ethics committee. The study was based on records review, re-verification, and re-analysis of routinely collected information, no patient identifiers were seen by the investigators, thus did not consent patients. This was consistent with the Uganda National Council for Science and Technology guidelines chapter 6.5; page 25 2007.

### Results

In the period from January 2018 to August 2020, a total of 14,275 contacts were traced for both PBC and non-PBC TB index patients at 48 facilities yielding 234 TB cases. Of 234 persons diagnosed with TB, 42.7% were contacts of non-PBC index patients. The proportion of males was higher than that of females for both groups diagnosed with TB who were contacts of non-PBC and PBC index patients (54.0% vs 50.8%) respectively. Both groups registered more adults (>15years) with TB as compared to children. The proportion of persons positive for HIV was higher among those contacts of non-PBC index patients diagnosed with TB (57.0% vs 24.6%). Apart from having higher proportion of contacts with fever among persons diagnosed with TB that were contacts of non-PBC index patients (80.0% vs 50.8%), the rest of the common symptoms for TB were nearly the same proportion among both groups (Table 1).

Although only 35% (4,919/14,036) of the contacts screened were contacts of non-PBC-index patients, they had a higher proportion of diagnosed TB patients. The proportion of contacts screened for TB was the same among contacts of non-PBC and PBC index TB patients (98.4% vs 98.3%) respectively. Overall, the TB screening yield was higher among contacts of non-PBC at 2.0%, compared to 1.4% among contacts of PBC index patients (Table 2). The TB screening yield was lower among contacts less than 15 years (1.5%) compared to 1.7% among contacts 15 years and above.

In terms of timing of TB diagnosis among contacts in relation to the index treatment initiation date, only 2(2%) of contact from non-PBC index patients indicated same day diagnosis compared to 17(14.9%) among contacts of PBC index patients. For both groups, over 80% of their contacts were diagnosed with TB within 3 months which is one month after the intensive TB treatment phase of the index case (Table 3).

On bivariate analysis, factors associated with diagnosing contacts as non PBC TB patients were: Type of TB of the index patient OR 1.72, 95% CI [1.0–2.9] p = 0.04; and age being young (under 15 years) OR 3.77, 95% CI [2.0–7.0] p = <0.05 (Table 4).

At multivariate logistic regression analysis, the only predictor for diagnosing contacts as non PBC TB patients was age under 15 years (adjusted odds ratio [aOR] 7.53, 95% CI [3.27–17.3] p = <0.05) (Table 4).

### Discussion

From our study, TB yield was 2.0% and 1.4% among close contacts to non-PBC and PBC index TB contacts. Similar findings in TB yield (1.7%) were observed in a multicentric study carried out among contacts to index TB patient in Kampala [9]. Relatively higher yield of 4% was obtained in a multicenter in America and Canada conducted among close contacts to

**Table 1. Characteristics of persons diagnosed with TB who were contacts of non-PBC and PBC index patients.**

| Characteristic | Category | Total (n = 234) | Contacts diagnosed with PBC TB patient (n = 137) | Contacts diagnosed as Non-PBC TB patient (n = 97) | P Value |
|---|---|---|---|---|---|
| Type of TB the index patient | **Index person** diagnosed as PBC-TB patient | 134(57.3) | 86(62.8) | 48(49.5) | **0.04** |
| | **Index person** diagnosed as Non-PBC-TB patient | 100(42.7) | 51(37.2) | 49(50.5) | |
| **Sex of Contact** | Female | 112(47.9) | 66(48.2) | 46(47.4) | 0.91 |
| | Male | 122(52.1) | 71(51.8) | 51(52.6) | |
| **Age of Contact** | 0–14 | 58(24.8) | 20(14.6) | 38(39.2) | <0.05 |
| | 15+ | 176(75.2) | 117(85.4) | 59(60.8) | |
| **HIV Status of contact** | Positive | 90(41.7) | 47(37.9) | 43(46.7) | 0.19 |
| | Negative | 126(58.3) | 77(62.1) | 49(53.3) | |
| **Relation of contact to index** | Household | 180(76.9) | 110(80.3) | 70(72.2) | 0.14 |
| | Co-worker | 54(23.1) | 27(19.7) | 27(26.8) | |
| **Contact with Cough** | Yes | 220(94.4) | 126(92.6) | 94(96.9) | 0.16 |
| | No | 13(5.6) | 10(7.4) | 3(3.1) | |
| **Contact with fever** | Yes | 148(63.3) | 87(63.5) | 61(62.9) | 0.92 |
| | No | 86(36.7) | 50(36.5) | 36(37.1) | |
| **Contact with weight loss** | Yes | 137(60.4) | 73(55.3) | 64(67.4) | 0.07 |
| | No | 90(39.6) | 59(44.7) | 31(32.6) | |
| **Contact with excessive night sweats** | Yes | 113(49.3) | 60(45.5) | 53(54.6) | 0.17 |
| | No | 116(50.7) | 72(54.5) | 44(45.4) | |

Missing data contact with weight loss and contact with excessive night sweats.

**Table 2. TB screening yield among contacts of non-PBC and PBC index TB patients.**

| Characteristic | Contacts Traced | Contacts Screened N (%) | Contacts with presumptive TB N (%) | Investigated contacts with presumptive TB N (%) | TB diagnosed among contacts with presumptive TB N (%) | TB classification category of the contact | | Screening Yield (%) |
|---|---|---|---|---|---|---|---|---|
| | N = 14,275 | N = 14,036 | N = 1,796 | N = 1,062 | N = 234 | PBC N = 137 | Non PBC N = 97 | 1.7 |
| **TB classification** | | | | | | | | |
| Non PBC-index patient | 4,999 | 4,919(98.4) | 563(11.4) | 382(67.8) | 100(26.1) | 51(51.0) | 49(49.0) | 2 |
| PBC-index patient | 9,276 | 9,117(98.3) | 1233(13.5) | 680(55.2) | 134(19.7) | 86(64.2) | 48(35.8) | 1.4 |
| **Age of Index** | | | | | | | | |
| 0–14 | 3,919 | 3,852(98.3) | 478(12.4) | 276(57.7) | 58(21.0) | 20(34.5) | 38(65.5) | 1.5 |
| 15+ | 10,356 | 10,184(98.3) | 1,318(12.9) | 786(59.6) | 76(22.4) | 117(66.5) | 59(33.5) | 1.7 |

**Table 3. Timing of TB diagnosis among contacts of non-PBC and PBC index patients.**

| Type of TB of Index patients | Contacts with documented timing of TB diagnosis | Month of diagnosis | | | | | |
|---|---|---|---|---|---|---|---|
| | | 1 month | 2 months | ≤3 months | >3 months | 3–5 months | >5 months |
| Non-PBC-Index patients | 97 | 56(57.7) | 14(14.4) | 78(80.4) | 19(19.6) | 16(16.6) | 11(11.3) |
| PBC-Index patients | 114 | 68(59.6) | 14(12.3) | 94(82.5) | 20(17.5) | 27(23.7) | 5(4.4) |

**Table 4. Logistic model of the characteristics associated with diagnosing contacts as non PBC TB patients.**

| Variable | Category | OR (95% CI) | P-Value | AOR (95% CI) | P-Value |
|---|---|---|---|---|---|
| **TB type of the index patient** | Non PBC | 1.72(1.01–2.92) | **0.04** | 1.50(0.80–2.82) | 0.20 |
| | PBC | 1 | | 1 | |
| **Age of contact** | 0–14 | 3.77(2.01–7.04) | **<0.05** | 7.53(3.27–17.3) | **<0.05** |
| | 15+ | 1 | | 1 | |
| **HIV Status of contact** | Positive | 1.44(0.83–2.48) | 0.19 | 1.44(0.71–2.93) | **0.31** |
| | Negative | 1 | | 1 | |
| **Contact with Cough** | Yes | 2.48(0.66–9.28) | 0.17 | 3.01(0.68–13.3) | **0.15** |
| | No | 1 | | 1 | |
| **Contact with weight loss** | Yes | 1.67(0.96–2.89) | 0.07 | 1.29(0.57–2.90) | **0.53** |
| | No | 1 | | 1 | |
| **Contact with excessive night sweats** | Yes | 1.44 (0.85–2.45) | 0.17 | 1.2(0.60–2.61) | **0.55** |
| | No | 1 | | 1 | |
| **Relation of contact to index** | Household | 0.63(0.34–1.17) | 0.15 | 0.56(0.26–1.18) | **0.15** |
| | Co-worker | 1 | | 1 | |

culture positive index TB cases [10]. In our study, TB screening among contacts was by symptom screening only and this could explain the lower yield compared to the American–Canadian study that combined symptom screening with Tuberculin Skin Test (TST) and X-ray when TST was positive. Since in the America–Canada study contacts were followed up annually for 4 years, it is possible that the contacts could have been exposed to TB from another source within the four-year period therefore giving the higher yield. In another study conducted in Kampala that combined symptom screening with TST and X ray, a much higher yield of 10% and 13% was found among first degree relatives and household contacts respectively [11].

Other sub-Saharan African country studies among pulmonary TB index patients produced a yield between 1.5% and 8%. Two studies in South Africa, one in North West Province targeting newly diagnosed TB index cases (≥15 years) had a yield of 7.8% [12] and another in a high-burden metropolitan district had a yield of 6.6% [13]. An Ethiopian study targeting all types of TB cases (≥18 years) had a yield of 6.5% [14] and a Tanzanian study targeting laboratory-confirmed TB cases (all ages) had a yield of 6.4% [15]. Another study in select facilities in Kampala showed overall yield for active TB of 1.7% from contacts. Several studies in countries with high TB incidence have shown that the prevalence may reach 5% or more among contacts. A prospective cohort study conducted in Uganda reported the detection of 6% of secondary cases among the household contacts of index TB cases.

These study findings indicated a higher percentage of TB among male contacts and contacts in the ages under 15 years. Weight loss, fever and cough were among the most common TB symptoms identified among contacts. These findings have been elicited in other studies that have also shown a higher percentage of TB among men and close contacts, coughing identified as one of the most common TB symptom among contacts diagnosed with TB and more TB in the ages under 15 years [10, 16, 17].

In this study we could not easily establish the exact date of diagnosis of TB in the index case from the secondary data we accessed. However, since infectivity reduces with start of TB treatment, we considered start of treatment as a reference point for timing of TB diagnosis among contacts. No significant difference in the timing of TB diagnosis among contacts was observed in the two groups of the index TB patients (non-PBC vs PBC). Most of the contacts (80% non-PBC and 82% PBC) were diagnosed with TB within the first 3 months of the index case's start

of treatment. This finding was not much different from the 82% of contacts diagnosed with TB within 3 months of diagnosis of the index case reported in a multicentric study conducted in America and Canada [10].

The evaluation had several strengths. First, data from this evaluation was collected routinely as part of a large on-going TB implementation project and thus fully reflects programmatic conditions in low resource and high burden settings. Findings from this study are therefore likely to be generalizable to similar settings. Secondly, our analysis adds to a growing body of literature on TB case finding that seeks to understand the yield of TB among contacts of PBC and non-PBC index TB patients, thus improving case-finding in routine healthcare settings.

Our study had some limitations. The secondary data accessed lacked information on date of TB diagnosis among the index TB patients. However, we used the TB treatment start date of the index case as a proxy variable to determine the timing of TB diagnosis. TB screening among the contacts was by symptom screening without TST. This could have resulted into some cases being missed thus leading to under estimation of the incident TB cases. A significant proportion of TB contacts with presumptive TB were not investigated due to missing data which could lead to potential information bias. This might also impact on the findings for the TB yield.

## Conclusion

The TB yield among contacts of non-PBC index cases is like that of contacts of PBC index cases and there is equally as much TB among the contacts to PBC index TB patients as is for contacts to non-PBC index TB patients. Irrespective of the type of TB (non-PBC vs PBC) of the index TB case, most contacts were diagnosed with TB disease within the first 3 months of the index case's start of TB treatment. The type of TB (PBC, non-PBC) disease diagnosed in the contacts is not dependent on the type of TB of their contact index TB patients. To improve TB case-finding, emphasis should be placed on contact investigation for household and close contacts of all other index cases with pulmonary TB regardless of it being PBC or non-PBC.

## Supporting information

**S1 File. Manuscript data.**
(XLSX)

## Acknowledgments

We acknowledge the support of the NTLP, and the health facility teams where this work was done. Our thanks also go to Defeat TB Project for partner coordination and technical and logistical support throughout the implementation period; the civil society organizations (CSOs); and community support teams. Katherine Fatta edited the paper.

## Author Contributions

**Conceptualization:** Herbert Kisamba, Abel Nkolo.

**Data curation:** Nicholas Sebuliba Kirirabwa.

**Formal analysis:** Herbert Kisamba, Nicholas Sebuliba Kirirabwa, Abel Nkolo.

**Funding acquisition:** Seyoum Dejene, Abel Nkolo.

**Investigation:** Herbert Kisamba, Nicholas Sebuliba Kirirabwa, Kenneth Mutesasira, Abel Nkolo.

**Methodology:** Herbert Kisamba, Nicholas Sebuliba Kirirabwa, Abel Nkolo.

**Supervision:** Herbert Kisamba, Kenneth Mutesasira, Seyoum Dejene.

**Validation:** Abel Nkolo.

**Writing – original draft:** Herbert Kisamba, Nicholas Sebuliba Kirirabwa, Kenneth Mutesasira.

**Writing – review & editing:** Herbert Kisamba, Nicholas Sebuliba Kirirabwa, Kenneth Mutesasira, Seyoum Dejene, Abel Nkolo.

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
