## [Decision Letter · Decision Letter 0]

12 Aug 2022

PONE-D-22-15242Tuberculosis yield among Contacts of non-Pulmonary Bacteriologically Confirmed Index TB patients in the urban setting of Central UgandaPLOS ONE

Dear Dr. Nicholas Sebuliba Kirirabwa,

Thank you for submitting your manuscript to PLOS ONE. After careful consideration, we feel that it has merit but does not fully meet PLOS ONE’s publication criteria as it currently stands. Therefore, we invite you to submit a revised version of the manuscript that addresses the points raised during the review process.

We look forward to receiving your revised manuscript.

Kind regards,

Sebsibe Tadesse, PhD

Academic Editor

PLOS ONE

Journal Requirements:

Reviewers' comments:

**Comments to the Author**

1. Is the manuscript technically sound, and do the data support the conclusions?

Reviewer #1: Yes

Reviewer #2: Yes

2. Has the statistical analysis been performed appropriately and rigorously? 

Reviewer #1: Yes

Reviewer #2: Yes

3. Have the authors made all data underlying the findings in their manuscript fully available?

Reviewer #1: Yes

Reviewer #2: Yes

4. Is the manuscript presented in an intelligible fashion and written in standard English?

Reviewer #1: Yes

Reviewer #2: Yes

5. Review Comments to the Author

Reviewer #1: Contact investigation is an important strategy for TB case finding

Abstract

- Correct the spelling for Abstract (section title)

- Line 41: Conclusion - Replace “developed” with “diagnosed” as the study is not able to determine this.

Background

- Line 59 -64: The authors mention ACF tool kit but it is not clear how this is related to contact investigation in this context.

Study setting

- The authors need to provide readers with the criteria used for non PBC diagnosis (including PCD) in their setting. This is not listed

Study procedures

- The authors list that data was abstracted from the health national TB contact investigation registers at the health facilities (line 88 – 90) however in the next section on data management and analysis, they state that data was imported from the project electronic database (line 96-97). Were there two data sources for this work? This might confuse the reader.

- The authors

Data management and analysis

- Line 99-100: The authors list “….TB diagnostic cascade, we calculated the proportion of presumptive TB (a patient who presents with symptoms or signs suggestive of TB identified out of all contacts traced”. Consider rephrasing the numerator as “a contact who has symptoms or signs suggestive of TB” vs a patient who presents……

- Line 101: Replace “Presumptive TB contacts” with “contacts with presumptive TB” (including throughout the text) as the former could mean something else

Results

- Line 116 to 117: The statement is not clear and might be confusing to the readers

- Table 1: The table is not clear. The authors list TB classification of index case with rows indicating PBC and non PBC under “characteristic” and then on the other hand they also have two columns indicating PBC and non PBC. What is the difference? The table title refers to characteristics of persons diagnosed with TB who were contacts …….. The authors need to clearly indicate what they are referring to. The column heading should be very clear

- Line 130 – 131: It would be better to have this statement as the introductory statement under the results section. This would help the reader to better understand where the figure 234 comes from.

- Line 131 – 134: The statement is not clear and might be confusing to the readers

- Table 2:

(i) Better formatting of the column headings

(ii) The column headings are not clear

(iii) Replace “Presumptive TB contacts” with “Contacts with presumptive TB”

(iv) Replace “Presumptive TB diagnosed” with “TB diagnosed among contacts with presumptive TB”

- It is important to authors to note that not all TB contacts identified were investigated – Fewer were investigated for PBC-index cases. This is not captured in their discussion

- Table 3:

(i) The column header “Documented timing….” Is not clear. What do the figures 97 and 114 refer to?

- Table 4

(i) The information from table 4 is not clear. What does “type of TB” refer to in this context. This comment also applies to the abstract – What do the authors mean when they state that age < 15 years is associated with type of TB? The authors would need to be more specific.

Discussion

- Study limitations: The authors did not consider the fact that significant proportions of TB contacts with presumptive TB were not investigated and yet this could be a key limitation for this study. Was this part of missing data? There is potential for information bias. This might also impact on the findings for the TB yield.

Reviewer #2: I have reviewed manuscript by the author : Nicholas Sebuliba Kirirabwa et al which is addressing a very important component of the End TB strategy. Recent studies have reveal transmission of TB by patients who are smear negative and this builds on that observation from real-life experience. Although this is a very important manuscript, I have highlighted the areas of improvement below for the authors.

Minor revisions;

1) a)The word " non-pulmonary bacteriologically confirmed TB" I not clear. Does it include Extra pulmonary TB? if not, may be better to use " Pulmonary bacteriologically negative TB patients"

b) Did the Non PBC have any additional tests before this classification? could these be added under population section?

2) Abstract L23: please write in past tense this this is work already done.

3) Abstract L26 specify the number of health facilities

4) L31-34 please include n(%)

5) L35 diagnosed with TB with in 3 months ... of the index TB diagnosis?

6) L37 ...associated with type of TB diagnosed from contact. please specify the type of TB which was the comparator to the reference.

7) L40 and throughout the manuscript, please avoid starting the sentence with an abbreviation.

8) L41 This sentence on this line is missed in the method section including the design

9) L68 please italicize all scientific names

10) Under methods L106: why did the author decide to use <=0.05 for inclusion in the multivariate analysis? usually a lower P-value at bivariate is usually used say <=0.2 to cater for interacting factors and biologically plausible factors such as HIV

11) L115 (54.0% vs 50.8)%) add respectively

12) L141 please add n(%)

13) L147 could the author repeat the analysis with factors having P-value <=0.2 added to the model?

14) L161 The word statistically significant characteristics is misleading. Kindly revise with a non-interpreted title for the table.

15) L206 .....cases being missed: could this be under or over estimation of the incident TB cases?

6. PLOS authors have the option to publish the peer review history of their article (what does this mean?). If published, this will include your full peer review and any attached files.

Reviewer #1: No

Reviewer #2: **Yes: **Willy Ssengooba

---

## [Author Response · Author response to Decision Letter 0]

19 Nov 2022

November 17, 2022

To

The Editorial Office, 

PLOS ONE

Dear Sebsibe Tadesse,

Re: Tuberculosis yield among Contacts of non-Pulmonary Bacteriologically Confirmed Index TB patients in the urban setting of Central Uganda ( PONE-D-22-15242.)

Thank you for your email dated 12th August 2022. We thank you and the reviewers for the time taken to read through our manuscript and the necessary suggestions made to its revision. We very much look forward to a final decision on the publication of our manuscript with your journal subject to adequate revision and response to comments raised by the reviewer. Based on the instructions provided in your email, we uploaded the file of the revised manuscript on the journal’s website. 

We considered and responded to all comments from the reviewers. Appended to this letter our point-by-point response. We, again, express our sincere gratitude to the reviewers who identified areas of our manuscript that needed modification. 

We would also like to thank you for allowing us to resubmit a revised copy of the manuscript and have prioritized making these very necessary changes to avoid any delays. 

We hope that in the revised manuscript, we have addressed all comments satisfactorily to be acceptable for publication in PLOS ONE. 

Sincerely,

Nicholas Kirirabwa Ssebuliba, MPH

Corresponding author

Knowledge Management and Program Coordination Manager

URC

Email: ksnicky@yahoo.co.uk.

Response to comments from Reviewers

To appropriately mark where changes were made in reference to the responses below, please consider reviewing the manuscript while “displaying the tracked changes/showing comments”.

Reviewer reports:

Reviewer #1: Contact investigation is an important strategy for TB case finding

Abstract

- Correct the spelling for Abstract (section title)

Thank you, we have corrected line 17

- Line 41: Conclusion - Replace “developed” with “diagnosed” as the study is not able to determine this.

This has been done. Line 42

Background

- Line 59 -64: The authors mention ACF tool kit but it is not clear how this is related to contact investigation in this context.

We have mentioned how the ACF tool kit relates to contact investigation line 65-66

Study setting

- The authors need to provide readers with the criteria used for non PBC diagnosis (including PCD) in their setting. This is not listed

This is very important, the criteria used for non PBC has been added line 88 -94

Study procedures

- The authors list that data was abstracted from the health national TB contact investigation registers at the health facilities (line 88 – 90) however in the next section on data management and analysis, they state that data was imported from the project electronic database (line 96-97). Were there two data sources for this work? This might confuse the reader.

Sorry for this confusion, we had one data source the project electronic database. We have cleared this on line 98.

- The authors

Data management and analysis

- Line 99-100: The authors list “….TB diagnostic cascade, we calculated the proportion of presumptive TB (a patient who presents with symptoms or signs suggestive of TB identified out of all contacts traced”. Consider rephrasing the numerator as “a contact who has symptoms or signs suggestive of TB” vs a patient 

Thank you for this comment, we have revised it appropriately line 109-110

- Line 101: Replace “Presumptive TB contacts” with “contacts with presumptive TB” (including throughout the text) as the former could mean something else

Thank you for this comment, we have revised it appropriately line 111-112

Results

- Line 116 to 117: The statement is not clear and might be confusing to the readers

Thank you for this comment, we have addressed this line 129-130

- Table 1: The table is not clear. The authors list TB classification of index case with rows indicating PBC and non PBC under “characteristic” and then on the other hand they also have two columns indicating PBC and non PBC. What is the difference? The table title refers to characteristics of persons diagnosed with TB who were contacts …….. The authors need to clearly indicate what they are referring to. The column heading should be very clear

Thank you for this comment, we have addressed

- Line 130 – 131: It would be better to have this statement as the introductory statement under the results section. This would help the reader to better understand where the figure 234 comes from.

Thank you, the statement has been moved to make the introductory statement line 124-125

- Line 131 – 134: The statement is not clear and might be confusing to the readers

We have addressed, this line 144-146

- Table 2:

(i) Better formatting of the column headings

(ii) The column headings are not clear

(iii) Replace “Presumptive TB contacts” with “Contacts with presumptive TB”

(iv) Replace “Presumptive TB diagnosed” with “TB diagnosed among contacts with presumptive TB”

Comments on Table 2 have been addressed

- It is important to authors to note that not all TB contacts identified were investigated – Fewer were investigated for PBC-index cases. This is not captured in their discussion

Thank you for this, We have added this in the discussion line 220-222

- Table 3:

(i) The column header “Documented timing….” Is not clear. What do the figures 97 and 114 refer to?

This has been addressed line 158

- Table 4

(i) The information from table 4 is not clear. What does “type of TB” refer to in this context. This comment also applies to the abstract – What do the authors mean when they state that age < 15 years is associated with type of TB? The authors would need to be more specific.

This has been addressed line 164 and line 38

Discussion

- Study limitations: The authors did not consider the fact that significant proportions of TB contacts with presumptive TB were not investigated and yet this could be a key limitation for this study. Was this part of missing data? There is potential for information bias. This might also impact on the findings for the TB yield.

Thank you for this, we have added this in the discussion line 220-222

Reviewer #2: I have reviewed manuscript by the author : Nicholas Sebuliba Kirirabwa et al which is addressing a very important component of the End TB strategy. Recent studies have reveal transmission of TB by patients who are smear negative and this builds on that observation from real-life experience. Although this is a very important manuscript, I have highlighted the areas of improvement below for the authors.

Minor revisions;

1) a)The word " non-pulmonary bacteriologically confirmed TB" I not clear. Does it include Extra pulmonary TB? if not, may be better to use " Pulmonary bacteriologically negative TB patients"

Thanks, this has been addressed line 90-94

b) Did the Non PBC have any additional tests before this classification? could these be added under population section?

No addition tests were done. This has been addressed under the study setting section line 90-94

2) Abstract L23: please write in past tense this this is work already done.

This is addressed line 23

3) Abstract L26 specify the number of health facilities

Added, line 26

4) L31-34 please include n(%)

These have been included line 32-34

5) L35 diagnosed with TB within 3 months ... of the index TB diagnosis?

This is addressed line 35

6) L37 ...associated with type of TB diagnosed from contact. please specify the type of TB which was the comparator to the reference.

This is addressed

7) L40 and throughout the manuscript, please avoid starting the sentence with an abbreviation.

Thank you, this was noted and addressed accordingly line 41

8) L41 This sentence on this line is missed in the method section including the design

This is addressed line104-107 

9) L68 please italicize all scientific names

Thank you, this was noted and addressed accordingly line 71

10) Under methods L106: why did the author decide to use <=0.05 for inclusion in the multivariate analysis? usually a lower P-value at bivariate is usually used say <=0.2 to cater for interacting factors and biologically plausible factors such as HIV.

This was a mistake we meant to use <=0.2, we have re-run the analysis based on this. see table 4

11) L115 (54.0% vs 50.8)%) add respectively

This has been addressed, line 128

12) L141 please add n(%)

This has been addressed, line 144

13) L147 could the author repeat the analysis with factors having P-value <=0.2 added to the model?

Thank you we have re-run the analysis based on this. see table 4 and line 160

14) L161 The word statistically significant characteristics is misleading. Kindly revise with a non-interpreted title for the table.

We revised this line 176

15) L206 .....cases being missed: could this be under or over estimation of the incident TB cases?

This has been addressed, line 223-224

---

## [Editor Report · Decision Letter 1]

28 Nov 2022

Tuberculosis yield among Contacts of non-Pulmonary Bacteriologically Confirmed Index TB patients in the urban setting of Central Uganda

PONE-D-22-15242R1

Dear Dr.Nicholas Sebuliba Kirirabwa,

We’re pleased to inform you that your manuscript has been judged scientifically suitable for publication and will be formally accepted for publication once it meets all outstanding technical requirements.

Kind regards,

Sebsibe Tadesse, PhD

Academic Editor

PLOS ONE

---

## [Editor Report · Acceptance letter]

12 Dec 2022

PONE-D-22-15242R1 

Tuberculosis yield among Contacts of non-Pulmonary Bacteriologically Confirmed Index TB patients in the urban setting of Central Uganda 

Dear Dr. Kirirabwa:

I'm pleased to inform you that your manuscript has been deemed suitable for publication in PLOS ONE. Congratulations! Your manuscript is now with our production department. 

Kind regards, 

on behalf of

Dr. Sebsibe Tadesse 

Academic Editor

PLOS ONE